# Synchronous Ovarian Sertoli–Leydig Cell and Clear Cell Papillary Renal Cell Tumors: A Rare Case Without Mutations in Cancer-Associated Genes

**DOI:** 10.3390/curroncol32080429

**Published:** 2025-07-30

**Authors:** Manuela Macera, Simone Morra, Mario Ascione, Daniela Terracciano, Monica Ianniello, Giovanni Savarese, Carlo Alviggi, Giuseppe Bifulco, Nicola Longo, Annamaria Colao, Paola Ungaro, Paolo Emidio Macchia

**Affiliations:** 1Dipartimento di Medicina Clinica e Chirurgia, Scuola di Medicina e Chirurgia, Università degli Studi di Napoli Federico II, 80131 Napoli, Italy; manuelamaceramd@gmail.com (M.M.); colao@unina.it (A.C.); 2Dipartimento di Neuroscienze e Scienze Riproduttive ed Odontostomatologiche, Scuola di Medicina e Chirurgia, Università degli Studi di Napoli Federico II, 80131 Napoli, Italy; simonemorra@outlook.com (S.M.); nicola.longo@unina.it (N.L.); 3Dipartimento di Sanità Pubblica, Scuola di Medicina e Chirurgia, Università degli Studi di Napoli Federico II, 80131 Napoli, Italy; marioascione1995@gmail.com (M.A.); carlo.alviggi@unina.it (C.A.); giuseppe.bifulco@unina.it (G.B.); 4Dipartimento di Scienze Mediche Traslazionali, Scuola di Medicina e Chirurgia, Università degli Studi di Napoli Federico II, 80131 Napoli, Italy; daniela.terracciano@unina.it; 5AMES, Centro Polidiagnostico Strumentale SRL, Casalnuovo di Napoli, 80013 Napoli, Italy; monica.ianniello@centroames.it (M.I.); giovanni.savarese@centroames.it (G.S.); 6Istituto degli Endotipi in Oncologia, Metabolismo e Immunologia “G. Salvatore” del CNR (IEOMI-CNR), 80131 Napoli, Italy

**Keywords:** Sertoli-Leydig cell tumor (SLCT), clear cell papillary renal cell carcinoma (CCP-RCC), androgenic ovarian tumor, genetic predisposition, case report

## Abstract

Sertoli–Leydig cell tumors of the ovaries (SLCTs) are rare androgen-producing neoplasms. Clear cell papillary renal cell carcinoma (CCP-RCC) is another rare tumor type that occurs in the kidney. We describe a unique case of a woman who was diagnosed with both tumors simultaneously. Both tumors were successfully removed by surgery. Despite extensive genetic testing, including DICER1 mutation screening and analysis of more than 280 cancer-related genes, no mutations were detected, suggesting that these tumors may have arisen by chance or unknown mechanisms. This case underlines the importance of a multidisciplinary approach and genetic screening, even if no mutations were detected. It also raises new questions about how such rare tumors may be linked, highlighting the need for further research and long-term surveillance.

## 1. Introduction

Sertoli–Leydig cell tumors of the ovary (SLCTs) are rare tumors of the reproductive tract that account for less than 0.5% of ovarian tumors and originate from Sertoli stromal cells [1,2,3,4]. SLCTs predominantly affect young women of reproductive age, with an average age at diagnosis of 25 years, but can occur at any age, including childhood and postmenopausal age [5,6,7].

SLCTs are often hormonally active, resulting in elevated serum androgen levels [7,8], and they are indeed the most common virilizing ovarian neoplasm [9]. Clinical presentation is highly variable, ranging from incidental findings on imaging studies to symptoms related to hormone production, such as virilization, menstrual irregularities, or abdominal pain due to tumor mass effect [7].

The tumors are typically unilateral, rarely (<1.5%) occurring bilaterally, and are generally confined to the ovary [9]. While the majority of SLCTs are benign and have an excellent prognosis after surgical resection, a subset of tumors may exhibit malignant behavior [4]. Several prognostic factors have been proposed to predict the malignant potential of SLCTs, including tumor size, histologic grade, mitotic index, and the presence of heterologous elements [10].

Recent studies have linked SLCTs to DICER1 syndrome, a rare genetic disorder that predisposes patients to a variety of tumors. DICER1 mutations, particularly those affecting the RNase IIIb domain, have been identified in approximately 60% of SLCTs, especially those with intermediate or poor histologic differentiation. In contrast, well-differentiated SLCTs are generally not associated with DICER1 alterations, suggesting possible differences in tumorigenic pathways [3,11,12,13]. Given the significance of these findings, genetic testing has become an essential part of the diagnostic workup of patients with SLCT.

In the age of precision medicine, genomic analysis plays a central role in clarifying the pathogenesis of rare tumors and adapting therapeutic strategies. However, the lack of detectable mutations in some patients complicates diagnosis and treatment. Case-based genomic studies, especially in rare tumor combinations, are valuable for exploring potential common molecular mechanisms and improving follow-up protocols.

Herein, we present a unique case of a 40-year-old woman diagnosed with pure ovarian SLCT and synchronous clear cell papillary renal cell carcinoma (CCP-RCC) with no detectable genomic DNA alterations. This case highlights the diagnostic challenges and the need to consider rare tumor associations and possible underlying genetic predispositions. It also highlights the importance of comprehensive diagnostic strategies, including hormonal, radiologic, and genetic testing, to ensure effective management and treatment.

## 2. Materials and Methods

### 2.1. Case Presentation

A 40-year-old Caucasian woman presented to the endocrinology outpatient clinic of AOU Federico II University Hospital with a 12-year history of secondary amenorrhea and progressive hirsutism affecting the face, chest, and limbs. Her medical history included hypertension, dyslipidemia, stage II obesity, goiter, and bilateral non-secreting adrenocortical adenomas. There was no known family history of malignancy.

Upon physical examination, the patient showed signs of virilization, including clitoromegaly, deepening of the voice, hair loss, and significant hirsutism quantified using the modified Ferriman–Gallwey scale [14] with a score of 27 points, indicating severe hirsutism.

The patient signed an informed consent form agreeing to the publication of this case report.

### 2.2. Serum Assays

Serum tests were performed in the clinical lab of the Federico II University Hospital. Cortisol, total testosterone, estradiol, progesterone, LH, FSH, ACTH, AFP, Ca19-9, Ca15-3, Ca125, ß-HCG, and CEA levels were measured by chemiluminescent immunoassays (CLIAs) (Advia Centaur or Centaur XP, Siemens Healthcare Diagnostics, Tarrytown, NY, USA). Serum DHEAS and androstenedione levels were measured by CLIA (Immulite 2000, Siemens Healthcare Diagnostics, Tarrytown, NY, USA). Serum 17OHP levels were assayed by ELISA (MG12181: Tecan IBL GmbH, Hamburg, Germany).

## 3. Results

### 3.1. Diagnosis

Hormonal tests (Table 1) revealed significantly elevated serum testosterone (518.9 ng/dL) and androstenedione levels (>10 ng/mL), indicating androgen-producing tumor activity. Dehydroepiandrosterone sulfate (DHEAS) and 17-hydroxyprogesterone (17-OHP) levels were within normal limits, ruling out androgen excess in the adrenal gland.

Serum levels of tumor markers CA 125, CA 19-9, CA 15-3, carcinoembryonic antigen (CEA), alpha-fetoprotein (AFP), and β-human chorionic gonadotropin (β-hCG) were within normal limits (Table 1).

Following the hormone tests, an imaging examination was essential to further characterize the adnexal mass. Ultrasonography remains the preferred method for the evaluation of ovarian lesions due to its safety profile, diagnostic efficacy, and ability to raise early suspicion when performed by personnel trained in gynecologic oncology. Although rare, Sertoli–Leydig cell tumors (SLCTs) typically appear as unilateral solid or solid–cystic adnexal lesions with sharp borders and abundant vascular flow during Doppler examination. While these features are not specific, they can help distinguish SLCTs from the more common epithelial tumors of the ovary [15].

In patients for whom transvaginal ultrasound is not an option, transrectal scanning may be a valid alternative to improve pelvic visualization. However, our patient refused both transvaginal and transrectal ultrasound examinations, so only a transabdominal pelvic ultrasound could be performed. Its diagnostic yield was limited due to the patient’s body habitus, so magnetic resonance imaging (MRI) of the abdomen and pelvis was performed. The MRI showed a 57 × 50 mm heterogeneous mass in the left ovary with contrast enhancement. In addition, a complex cystic lesion up to 44 mm in size was found in the upper pole of the right kidney.

A complementary contrast-enhanced computed tomography scan of the abdomen/pelvis described the left adrenal mass as a large, solid, polylobular lesion measuring 58 × 46 mm, with a discrete serous fluid component and heterogeneous post-contrast enhancement. The right renal lesion was described as a predominantly exophytic, cystic, nodular formation measuring 46 × 38 mm, with a hypervascularized component in the peripheral part.

To assess adrenal function, a stimulation test with 250 µg of IM adrenocorticotropic hormone (ACTH; Synacthen 0.25 mg/mL, Alfasigma, Bologna, Italy) was performed. The adrenal response to the ACTH test was normal, with cortisol and 17-hydroxyprogesterone levels increasing within the expected ranges. DHEA-S levels increased only slightly after 30 and 60 min. Finally, the concentrations of testosterone and androstenedione were pathologically elevated both at the beginning and during the test. The normal responses of cortisol and 17-hydroxyprogesterone confirmed intact adrenal function. However, the attenuated DHEA-S response, together with the persistently high testosterone and androstenedione levels, indicated a non-adrenal source of hyperandrogenism, strongly suggesting an ovarian origin (Table 2).

Subsequently, a gonadotropin-releasing hormone (GnRH) stimulation test was performed with the subcutaneous administration of 0.1 mg buserelin acetate (BBFarma, Samarate, VR), preceded by adrenal suppression with 1 mg dexamethasone (Decadron, Savio Pharma Italia S.r.l., Pomezia, RM) administered at 11 pm the night before the test to assess ovarian function in the absence of adrenal androgens [16]. The results are shown in Table 3. Despite the dexamethasone-induced suppression of adrenal androgens, the patient showed a markedly increased androstenedione response after GnRH stimulation. This finding confirmed the ovarian responsiveness to gonadotropins and indicated a steroidogenically active ovarian tumor.

### 3.2. Surgical Treatment

The patient underwent a robot-assisted laparoscopic left salpingo-oophorectomy. Histopathological examination of the ovarian mass confirmed a well-differentiated 50 × 43 × 36 mm SLCT with a solid cystic structure and a hemorrhagic brown appearance, displacing the entire ovary.

Microscopically, the neoplasm consisted of medium-sized cells with round nuclei and abundant eosinophilic cytoplasm, with a minimal presence of clear cells. There was no evidence of capsular invasion. Immunohistochemical analysis revealed positive staining for Calretinin, WT1, Inhibin, and focal positivity for MART-1, with a Ki67 proliferation index of 7%.

In the same surgical session, the patient underwent robotic laparoscopic partial nephrectomy of the right kidney (Figure 1b). Histopathologic analysis identified the lesion as CCP-RCC, grade 2. The tumor was 47 × 45 × 31 mm, well circumscribed, and composed of papillae and nests of clear cells with pleomorphic nuclei. Immunohistochemical staining was positive for CK7 and showed “patchy” positivity for CD10, with negative staining for GATA3. No capsular invasion was noted, although focal necrosis was present.

### 3.3. Genetic Studies

Given the association of SLCTs with DICER1 syndrome, genetic testing was performed to verify the presence of mutations in the DICER1 gene. Genomic DNA was extracted from the patient’s peripheral blood using the MagCore Nucleic Acid Extraction Kit (Diatech Pharmacogenetics, Ancona, Italy). Whole exome sequencing (WES) was performed using Kapa HyperPlus kits (Roche Molecular Systems Inc., Santa Clara, CA, USA) and sequencing took place on a NovaSeq 6000 platform (Illumina Inc., San Diego, CA, USA) at an average depth of at least 100× (see Appendix A for more details).

Complementary to this analysis, multiplex ligation-dependent probe amplification (MLPA) was performed to detect any deletions or duplications within the DICER1 gene (see Appendix A for details).

Both WES and MLPA analysis were negative for point mutations, frameshift mutations, and copy number variations in the DICER1 gene, ruling out a DICER1-related genetic predisposition.

In addition, a panel of approximately 280 genes potentially associated with hereditary and sporadic cancers was screened for mutations using high-throughput next-generation sequencing (NGS) (Appendix A This approach allows the identification of pathogenic or potentially pathogenic mutations in 8–15% of cases and increases the diagnostic, prognostic, and therapeutic value of next-generation sequencing in oncology (details in the Appendix A). No mutations were detected in any of the other genes analyzed, with the exception of a single variant (c.1814C > G p.(Thr605Ser, rs587781616) exon: 4/10) in the MSH6 gene (NM_000179.2). MSH6 is listed as a disease gene for Lynch syndrome and familial endometrial cancer, both of which are diseases with autosomal dominant transmission. The variant is present in the dbSNP database and is annotated in ClinVar as a variant with a conflicting interpretation of pathogenicity, and it is described 9/10 times as a variant of undetermined significance (VUS) for Lynch syndrome and hereditary cancer predisposition.

An overview of the genomic testing methods, their results, and their clinical relevance is summarized in Table 4.

## 4. Discussion

This fascinating case highlights the diagnostic and therapeutic challenges of SLCTs, especially when they co-occur with other rare neoplasms such as CCP-RCC. Although SLCTs are rare, they should be considered in women with symptoms of virilization. Our patient presented with classic signs of androgen excess, which led to a comprehensive diagnostic workup. Initial imaging revealed lesions in both the ovary and kidney. The diagnosis was confirmed by histopathology, which remains the definitive method for diagnosing ovarian tumors. The robotic laparoscopic approach allowed for complete resection of the tumor, which is important not only for diagnosis, but also for optimal treatment outcomes, as SLCTs can be potentially malignant.

The simultaneous diagnosis of CCP-RCC is intriguing and raises questions about possible genetic links. Hereditary breast and ovarian cancer syndrome and Lynch syndrome are the most common hereditary syndromes associated with gynecologic tumors, but other forms of hereditary cancer can also occur. These forms are often associated with inherited gene mutations, and clinical detection of neoplasm associations has relevant clinical implications as it may help to reveal an undiagnosed syndrome and recommend genetic counseling and formal risk assessment to the patient [17]. SLCTs and renal tumors have recently been linked to DICER1 syndrome, an inherited cancer predisposition syndrome also known as DICER1-related pleuropulmonary blastoma cancer predisposition syndrome (OMIM 601200). The DICER1 gene (OMIM 606241) is located on 14q32.13 (hg18) and consists of 27 exons [18]. It encodes for the enzyme cytoplasmic endoribonuclease (RNase) III, which plays a central role in the RNA interference pathway. It cleaves double-stranded RNA molecules into small RNAs, including microRNA (miRNA) and small interfering RNA (siRNA). DICER1 facilitates the incorporation of these RNAs into the Argonoute protein to form the RNA-induced silencing complex (RISC) [19]. The activated RISC recognizes a specific mRNA target sequence and can either initiate the degradation of the molecule or inhibit its translation. DICER1 syndrome (OMIM#601200) is a rare pleiotropic tumor predisposition syndrome that is inherited in an autosomal dominant manner but can also occur de novo in the germline or in somatic mosaic form. It is estimated that 80% of pathogenic germline variants are inherited from one parent and 20% occur de novo [20]. In addition, mutations in DICER1 occur in various cancers, e.g., sporadic pleuropulmonary blastoma; gonadal, Wilms, and endometrial tumors; and anaplastic sarcomas of the kidney [21,22]. Multinodular goiter-1 (MNG1) with or without SLCT (OMIM 138800) is also caused by heterozygous mutations in the DICER1 gene. Most of the described mutations in DICER1 are point or frameshift mutations accompanied by loss of function, but deletions of the entire DICER1 locus and in- or out-of-frame intragenic DICER1 deletions have also been identified [23]. Our patient tested negative for both DICER1 mutations and deletions of this gene. Consistent with the current knowledge, no DICER1 mutations were found in our patient’s well-differentiated SLCT, which is a subtype generally not associated with DICER1 variants [3]. This finding suggests that the SLCT and CCP-RCC in this case may have occurred sporadically, as part of a hereditary cancer syndrome. However, the possibility that other genetic factors or signaling pathways contributed to the development of these tumors cannot be ruled out. To explore possible genetic causes, we also investigated a panel of approximately 280 genes known to be associated with various tumors using NGS. No additional mutations were identified, apart from a previously reported variant of unknown significance (VUS) in the MSH6 gene (NM_000179.2), which is listed in the dbSNP database and annotated in ClinVar with conflicting interpretations of pathogenicity.

One limitation of this study is the inability to thoroughly investigate the presence of somatic mutations in tumor tissues. Although both WES and MLPA of germline DNA excluded known DICER1 point mutations and copy number alterations, we acknowledge that deep intronic variants and somatic mosaicism—which are particularly relevant in moderately or poorly differentiated SLCTs—may not be excluded without sequencing the tumor tissue. We are aware of this limitation, but due to technical and logistical constraints, we were unable to perform the necessary analyses to investigate this aspect in depth, which could have provided further insight into the genetic alterations potentially driving tumor development and progression. In addition, the presence of epigenetic alterations that could predispose individuals to the development of multiple primary tumors has not been ruled out.

The prognosis of SLCT depends on tumor features like size, grade, and mitotic index. Most SLCTs have excellent outcomes, but some can be aggressive. Our patient’s well-differentiated tumor suggests a favorable prognosis; nevertheless, a regular follow-up is crucial, especially in the context of concurrent CCP-RCC.

## 5. Conclusions

This case report presents the rare synchronous occurrence of an ovarian SLCT and CCP-RCC without detectable DICER1 or other pathogenic germline mutations. To our knowledge, this is the first documented association of these two neoplasms. It underscores the importance of considering possible underlying genetic predispositions when studying patients with multiple rare tumors.

Although the renal lesion could be an incidental finding, the synchronous occurrence in a relatively young adult raises the possibility of common pathogenic mechanisms that warrant further molecular investigation. The favorable outcome after surgical resection of both tumors underscores the value of a multidisciplinary diagnostic and therapeutic approach that includes endocrinology, radiology, surgery, and pathology.

The absence of genomic alterations despite extensive testing suggests the possible involvement of epigenetic factors, somatic mutations not detected by current testing, or as yet unidentified hereditary cancer syndromes. These findings emphasize the need for further investigation of the molecular landscape of SLCTs and their possible associations. In similar cases, genetic counseling should be considered even if no mutations have been detected.

Finally, broader data collection through multicenter registries could improve the understanding of rare tumor combinations and facilitate the identification of novel genetic factors. Such efforts could ultimately inform both clinical surveillance strategies and future diagnostic guidelines.

## Figures and Tables

**Figure 1 curroncol-32-00429-f001:**
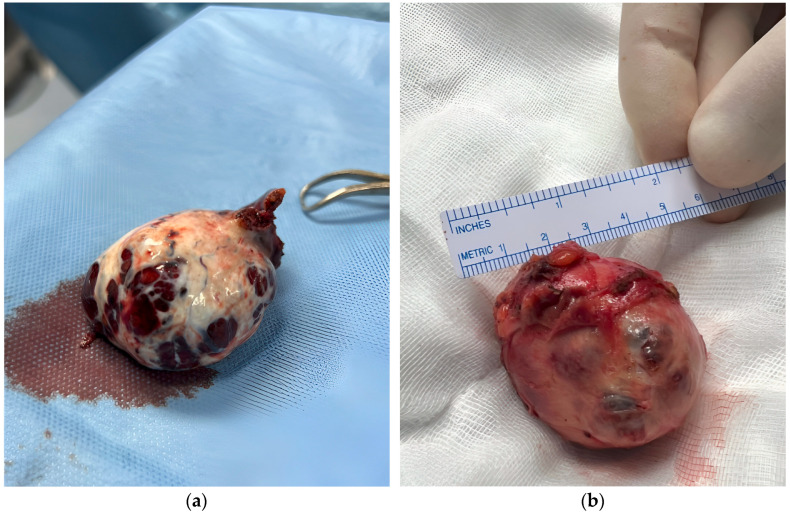
Surgical specimens after robot-assisted laparoscopy. Pictures show both left ovarian mass measuring 50 × 43 × 36 mm (**a**) and right kidney mass measuring 47 × 45 × 31 mm (**b**).

**Table 1 curroncol-32-00429-t001:** Basal hormone and tumoral markers levels in patient’s serum.

	Measured Value	Reference Range
Testosterone (ng/dL)	**518.9**	7.0–59.0
DHEA-S (µg/dL)	72.2	25.9–460.2
Androstenedione (ng/mL)	**>10**	1.0–4.5
FSH (mU/mL)	8.3	Follicular stage: 2.5–10.2 Ovulation peak: 3.4–33.4 Luteal stage: 1.5–9.1 Menopause: 23.0–116.3
LH (mU/mL)	5.4	Follicular stage: 1.9–12.5 Ovulation peak: 8.7–76.3 Luteal stage: 0.5–16.9 Menopause: 15.9–54.0
Progesterone (ng/mL)	1.2	Follicular stage: 0.2–1.40 Luteal stage: 3.4–28.0 Menopause: <0.73
Estradiol (pg/mL)	78.9	Follicular stage: 19.5–144.2 Ovulation peak: 63.9–356.7 Luteal stage: 55.8–214.2 Menopause: <32.2
ACTH (pg/mL)	7.0	10.0–130.0
Cortisol (µg/mL)	16.3	4.3–22.4
17OH-P (ng/mL)	3.39	0.35–4.13
CA 125 (U/mL)	21,8	<35
CA 19.9 (U/mL)	16.5	<35
CA 15-3 (U/mL)	14.0	<38.7
CEA (ng/mL)	1.3	Smokers: <4.9 Non-smokers: <2.5
AFP (ng/mL)	1.93	<10.0
ß-HCG (mU/mL)	<2.0	<4.0

Abnormal values are indicated in bold. DHEA-S: dehydroepiandrosterone sulfate; FSH: follicle-stimulating hormone; LH: luteinizing hormone; ACTH: adrenocorticotropin hormone; 17OH-P: 17-hydroxyprogesterone; CA: cancer antigen; CEA: carcinoembryonic antigen; AFP: human α-fetoprotein; β-hCG: β-human chorionic gonadotropin.

**Table 2 curroncol-32-00429-t002:** Adrenal hormone response to ACTH stimulation test.

	Reference Range	0′ (Basal)	30′	60′
Cortisol (μg/dL)	4.3–22.4	13.3	35.6	43.8
Testosterone (ng/dL)	7.0–59.0	500	508	512
Andronestedione (ng/mL)	1.0–4.5	>10	>10	>10
DHEA-S (μg/dL)	25.9–460.2	42.4	45	48.3
17 OH-P (ng/mL)	0.35–4.13	2.5	2.8	3.20

Adrenal hormone response to ACTH stimulation test. Test was performed by administering 250 µg of ACTH intramuscularly. Hormone levels were measured at baseline (0 min), as well as 30 min and 60 min after ACTH administration. Abbreviations: DHEA-S: dehydroepiandrosterone sulfate; 17-OHP: 17-hydroxyprogesterone.

**Table 3 curroncol-32-00429-t003:** FSH, LH, and androgen response to gonadotropin-releasing hormone (GnRH) stimulation test.

	0′ (Basal)	30′	60′	90′	120′
FSH (mU/mL)	3.6	7.0	8.4	18.2	26.0
LH (mU/mL)	2.3	10.8	15.6	48.5	67.2
Testosterone (ng/dL)	146.3	145.1	159.5	214.8	235.0
Androstenedione (ng/mL)	>10	>10	>10	>10	>10
DHEA-S (µg/mL)	37.0	38.0	41.0	40.0	40.0
17 OH-P(ng/mL)	1.75	1.76	3.20	6.48	11.28

GnRH stimulation test was performed by administering 0.1 mg of busereline acetate, preceded by overnight dexamethasone suppression test (1 mg). Values were measured at baseline and at 30, 60, 90, and 120 min. Abnormal values are indicated in bold. FSH: follicle-stimulating hormone; LH: luteinizing hormone; DHEA-S: dehydroepiandrosterone sulfate; 17-OHP: 17-hydroxyprogesterone.

**Table 4 curroncol-32-00429-t004:** Summary of genetic tests performed and key findings.

Genetic Test	Methodology	Result	Interpretation
*DICER1* sequencing	WES	No point mutations, frameshifts	DICER1 syndrome excluded
*DICER1* CNV analysis	MLPA	No deletions or duplications detected	
Hereditary cancer gene panel (~280 genes)	High-throughput NGS	One variant (c.1814C > G; p.Thr605Ser) in *MSH6* gene	VUS; no definitive hereditary syndrome identified

Abbreviations: WES: whole exome sequencing; CNV: copy number variation; MLPA: multiplex ligation-dependent probe amplification; NGS: next-generation sequencing; VUS: variant of uncertain significance.

## Data Availability

No new data were created or analyzed in this study. Data sharing is not applicable to this article.

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
