# Peer review of "Synchronous Ovarian Sertoli–Leydig Cell and Clear Cell Papillary Renal Cell Tumors: A Rare Case Without Mutations in Cancer-Associated Genes"

_curroncol, 2025, doi:10.3390/curroncol32080429_

Round 1

Reviewer 1 Report

Comments and Suggestions for Authors

Thank you for this interesting case report. A few comments: Well diff SLCTs are not generally associated with DICER1 variants. That said, somatic testing is needed to exclude this as available germline testing including that described in this report may miss intronic variants and some deletions. Mosaicism is also a consideration (again for moderately or poorly diff tumors more than well diff tumors based on available data). Also, the citations should be updated to include more recent reports of SLCT and well diff SLCT and the updated GeneReviews for DICER1. The patient photos are not needed/relevant and may unnecessarily violate confidentiality. Also, the comment regarding the patient's sexual history is not relevant to the report and both the identifiable photos of the patient and her sexual history should not be included for ethical reasons. 

Author Response

Comment 1: Well diff SLCTs are not generally associated with DICER1 variants.

Thank you for this comment. We have clarified this aspect both in the Introduction (lines 59-61) and Discussion (Lines 266-268).

Comment 2: Somatic testing is needed to exclude this as germline testing may miss intronic variants and some deletions. Mosaicism is also a consideration (again for moderately or poorly diff tumors more than well diff tumors based on available data).

We fully agree. We acknowledged this limitation explicitly in the revised Discussion (lines 277-280)"

Comment 3: Citations should be updated to include more recent reports of SLCT and the updated GeneReviews for DICER1.

We included recent references such as Lyu et al. (2025) and Nelson et al. (2025) in the Introduction. We also cited the GeneReviews article (Schultz et al., 2014) and have verified it remains the latest version available online. If a newer version becomes available before publication, we will be happy to update the citation.

Comment 4: The patient photos are not needed/relevant and may unnecessarily violate confidentiality.

The clinical photographs have been removed from the manuscript.

Comment 5: The comment regarding the patient's sexual history is not relevant to the report.

The phrase suggesting the patient's "virgin status" has been removed. The revised version neutrally discusses the refusal of transvaginal and transrectal ultrasound, without reference to sexual history (Line 124).

Reviewer 2 Report

Comments and Suggestions for Authors

It was a pleasure to read this case report. The Authors present a case of Sertoli Leydig cell tumor and clear cell papillary renal cell carcinoma. Hormonal evaluation, imaging studies and extensive genetic testing, including DICER1 mutation analysis and whole-exome sequencing were performed to diagnosis. Specifically the study of genetic alterions is the most interesting topic of the case, due to limited current evidence in this field.

Here I report my insight:
* Title “Synchronous” instead of “Concurrent”
* Abstract: well written, however I suggest to modify conclusion that could be more informative. It would be useful to explicitly state the negative result of DICER1 and WES/MLPA testing, which is relevant for the rarity of this case
* Introduction: Presentation of case is valid. Nonetheless, the discussion would expance from reference by Lyu et al. (Lyu Z, et al. Pathological and clinical insights into DICER1 hotspot mutated Sertoli-Leydig cell tumors: a comparative analysis. Diagn Pathol. 2025 Apr 29;20(1):55. doi: 10.1186/s13000-025-01657-8), which showed that DICER1 mutations are exclusively found in intermediate and poorly differentiated SLCTs, and not in well-differentiated subtypes like in this case. This would reinforce the genetic distinction implied by the authors’ findings
* Methods
CTH and GnRH stimulation protocols are described, but no interpretive comment is given in-text/results section
Modify “interoexternal part” (Line 116)
Somatic sequence of alteration of tissue was performed?
*Discussion
The discussion appear interesting and comprehensive from a genetic standpoint, I believe it would benefit from the implementation of a section addressing the role of preoperative imaging. In particular, ultrasound evaluation may be of critical importance in borderline cases, as it can support the diagnostic hypothesis and guide the most appropriate therapeutic strategy. Furthermore, I would like to suggest you the use of transrectal ultrasound in virgo patients, where transvaginal access is not feasible, since pelvic ultrasound by experienced sonographers in the field of gynaecological oncology allow an high chance of early diagnosis and personalized treatment. The patient's clinical features were highly suggestive of an androgen-producing neoplasm. Nevertheless, ultrasound remains an essential first-line tool in the evaluation of ovarian masses and can provide valuable preoperative information, especially in less subclinical presentations. Sertoli Leydig tumors appear on ultrasound as unilateral, solid or solid-cystic adnexal masses with well-defined margins. Doppler evaluation often reveals prominent vascularization. These features, although not pathognomonic, may aid in distinguishing SLCTs from more common epithelial tumors (Bruno M, et al. Sonographic characteristics of ovarian Leydig cell tumor. Ultrasound Obstet Gynecol. 2023 Sep;62(3):441-442. doi:10.1002/uog.26212).

Author Response

Comment 1: Title: use “Synchronous” instead of “Concurrent”

Thank you. The title has been corrected to use “Synchronous. The same correction has been done in the text of the manuscript

Comment 2: Abstract: conclusion should explicitly state the negative result of DICER1 and WES/MLPA testing.

We agree. The Abstract has been extensively revisited and additional details have been included.

Comment 3: Reference to Lyu et al. (2025) is important to reinforce the lack of DICER1 mutations in well-differentiated SLCTs.

Reference has been added (Ref. 13) and discussed in the Introduction to support the distinction between differentiated subtypes (Line 62).

Comment 4: CTH and GnRH stimulation protocols are described, but no interpretive comment is given.

Thank you for highlighting this important aspect. The Results section was updated to provide interpretation of both tests (lines 141-146 and 159-162).

Comment 5: Modify “interoexternal part” (Line 116)

Corrected. The text now reads: "a hypervascularized component in the peripheral part." (line 136)

Comment 6: Somatic sequence of tumor tissue performed?

We clarified that somatic testing was not performed due to technical/logistical limitations, and acknowledged this limitation explicitly in the Discussion (lines 277-280).

Comment 7: Imaging and ultrasound should be better addressed.

This point has been expanded in lines 114-127 and the suggested reference (Bruno et al. (2023) has been included) as well.

Comment 8: Patient photos and sexual history are not ethically appropriate.

Clinical images have been removed and references to sexual history eliminated, in full accordance with ethical standards.

Round 2

Reviewer 2 Report

Comments and Suggestions for Authors

The authors have addressed the reviewers’ comments and have improved the quality of the case report